# Demonstration of resonant tunneling effects in metal-double-insulator-metal (MI²M) diodes

Amina Belkadi [1✉], Ayendra Weerakkody[1] & Garret Moddel[1]

Although the effect of resonant tunneling in metal-double-insulator-metal (MI²M) diodes has been predicted for over two decades, no experimental demonstrations have been reported at the low voltages needed for energy harvesting rectenna applications. Using quantum-well engineering, we demonstrate the effects of resonant tunneling in a Ni/NiO/Al$_2$O$_3$/Cr/Au MI²M structures and achieve the usually mutually exclusive desired characteristics of low resistance ($R_0^{DC} \sim 13$ kΩ for 0.035 μm$^2$) and high responsivity ($\beta_0 = 0.5$ A $W^{-1}$) simultaneously. By varying the thickness of insulators to modify the depth and width of the MI²M quantum well, we show that resonant quasi-bound states can be reached at near zero-bias, where diodes self-bias when driven by antennas illuminated at 30 THz. We present an improvement in energy conversion efficiency by more than a factor of 100 over the current state-of-the-art, offering the possibility of engineering efficient energy harvesting rectennas.

[1] Department of Electrical, Computer and Energy Engineering, University of Colorado at Boulder, Boulder, CO, USA. ✉email: amina.belkadi@colorado.edu

Optical rectennas, combinations of micron-scale antennas and sub-micron diodes, provide a promising way to efficiently harvest low-grade waste heat. In our optical rectennas, we use metal–insulator–metal (MIM) diodes, which are compatible with standard CMOS fabrication processes and have a tunneling transit time of electrons through thin insulators (<5 nm) on the order of femtoseconds[1]. The challenge in designing MIM diodes suitable for high-frequency applications is in achieving low resistance and high responsivity simultaneously. A low resistance is important to obtain a low RC time constant and hence a high cutoff frequency. A high responsivity, defined as the DC current generated per unit power of incident radiation[2], is needed for high rectification efficiency. Schottky diodes have good rectification characteristics with responsivity values up to 5 A W$^{-1}$, but their high resistance results in a cut-off frequency of around 40 GHz[3,4]. Common methods of increasing MIM diode responsivity, such as increasing oxide thicknesses or barrier heights, result in an increase in resistance. This responsivity–resistance trade-off was experimentally observed by Bean et al. for single-insulator MIM diodes[5]. Herner et al. also observed this trend by fabricating and measuring hundreds of Co/Co$_3$O$_4$/TiO$_2$/Ti MI$^2$M diodes, where higher responsivity values came at the expense of higher resistance values[6]. Breaking this well-established trend requires the use of a non-standard approach, such as resonant tunneling, which has been predicted to offer a reduction in resistance with an increase in diode nonlinearity[7,8]. These two normally competing characteristics when achieved simultaneously could move energy harvesting rectennas from the exploration phase, where total power conversion efficiency is ~10$^{-10}$, to commercial phase, with a promise of orders of magnitude improvement in diode rectification efficiency. For almost two decades, numerous theoretical and experimental studies were performed to explore the possibility of achieving resonant tunneling in MIM structures to use in optical rectennas[7–12]. Because of the difficulty of achieving resonant tunneling at room temperature and near zero-bias, all experimental observations of resonant tunneling in MIM structures have been limited to high voltages (>1 V)[11–17], rendering the results unusable for energy harvesting where self-biasing occurs at ~100 μV. Demonstrating resonant tunneling at room temperature requires material engineering and careful properties control to observe this very sensitive effect. In this work, we present the first experimental demonstration of resonant tunneling effects in MI$^2$M diodes, where a reduction in resistance is observed with an increase in responsivity. Controlling oxide thicknesses to sub-nanometer accuracy has led to the observation of resonant tunneling close to zero-bias in Ni/NiO/Al$_2$O$_3$/Cr/Au diodes. We verified the possibility of extending the effects of resonant tunneling to high frequency by measuring these diodes in rectennas under 10.6 μm illumination, and achieving an improvement in overall conversion efficiency over structures without resonant tunneling. These MIM diodes present the best multi-terahertz current–voltage (I(V)) characteristics to date.

## Results

### Breaking the responsivity/resistance trend using resonant tunneling. 
Figure 1a shows the responsivity/resistance trade-off for simulated MI$^2$M diodes with the material set M$_1$/Ox$_1$/Ox$_2$/M$_2$, where M$_1$ and M$_2$ are the respective top and bottom metal electrodes of the diodes, and Ox$_1$ and Ox$_2$ are the two oxides of the diode. These simulated diodes (black dots) were based on diodes fabricated and measured by Herner et al., where a targeted thickness ratio of 1:1 for the two oxides resulted in a spread marked by the gray dashed line[6]. The data spread observed in fabrication across a wafer by Herner et al.[6] can be explained through simulations with a ±4 Å thickness variation in the two oxides, Ox$_1$ and Ox$_2$, due to non-uniformity of oxide growth and deposition over the wafer. The dominant tunneling mechanism in

such a structure is Fowler–Nordheim tunneling in Ox$_1$, where electrons tunnel through a part of energy barrier to the conduction band of Ox$_1$, and direct tunneling in Ox$_2$, where electrons tunnel through the whole energy barrier. Figure 1a also shows the effects of varying the thickness of Ox$_1$ over the range of 1–20 Å, presented as thickness sweep simulation, while maintaining the thickness of Ox$_2$ fixed at 1 nm. We expected responsivity to increase with resistance as oxide thickness and tunneling distance were increased, based on the hundreds of different diodes that we have fabricated and measured, as well as trends observed by Herner et al.[6] and Bean et al.[5]. Contrary to expectations, as the thickness of Ox$_1$ increased beyond a certain oxide ratio, resistance appeared to be held constant while responsivity continued to increase, as shown in Fig. 1a, where the thickness sweep line bends up. This puzzling trend had never before been predicted theoretically with variations in thickness or for diode figures of merit (responsivity and resistance) near zero bias. Varying the thickness of the first (Ox$_1$) allows for breaking the responsivity–resistance trend by crossing the dashed gray line, as seen in the inset in Fig. 1a.

Figure 1b shows a non-monotonic responsivity-resistance relationship at zero-bias for fabricated Ni/NiO/Al$_2$O$_3$/Cr/Au diodes. Each fabricated wafer was divided into four Ni/NiO/Al$_2$O$_3$/Cr/Au diodes batches with nominal NiO thicknesses of 3, 4, 5, and 6 nm, while maintaining the nominal thickness of Al$_2$O$_3$ at 1.3 nm. As shown from the relationship between mean values in Fig. 1b, responsivity increases with increasing thickness in the range of 3–5 nm (0.43 A W$^{-1}$ at 3 nm, 0.52 A W$^{-1}$ at 4 nm, and 0.59 A W$^{-1}$ at 5 nm), but resistance drops for the 4 nm structure before increasing for the 5 nm structure (10 kΩ at 3 nm, 4 kΩ at 4 nm, and 50 kΩ at 5 nm). Data presented includes diodes fabricated on three different wafers, in randomized orders (see the subsection "Device fabrication" in the "Methods" section).

### Simulation-based resonant tunneling analysis. 
Figure 2a shows the measured and simulated DC I(V) characteristics of an Ni/NiO/Al$_2$O$_3$/Cr/Au diode with an oxide thickness ratio of 4:1.3 nm. We used a quantum mechanical diode simulator that accounts for resonant tunneling in a multi-barrier structure to study our fabricated MI$^2$M diodes[7,8]. The simulator uses a transfer matrix method to solve a time-independent Schrödinger equation and calculate transmission amplitudes, with a Hamiltonian matrix constructed to determine the bound states in a quantum well (tunneling probabilities for the structures and the location of quasi-bound states in the triangular quantum well are discussed in Supplementary Note 1). The MI$^2$M diode areas (~0.035 μm$^2$) and nominal oxide layer thicknesses were measured using scanning electron microscopy (SEM) and variable-angle spectroscopic ellipsometry (VASE), respectively. Measured areas and oxide thicknesses varied by up to 11% and 12.5%, respectively. These measured values were used as a starting point for the simulations, and were varied, along with material properties such as barrier heights and effective mass, to achieve the best fit of the measured diode. The best fit was achieved using work functions of 5 and 4.47 eV for Ni and Cr/Au respectively, and electron affinity values of 4.7 and 3.45 eV for NiO and Al$_2$O$_3$, respectively. A table comparing these values to literature values is included in Supplementary Note 2. We extracted thicknesses of 3.2 and 1 nm for NiO and Al$_2$O$_3$, respectively. Electrical thicknesses and nominal thicknesses tend to differ as seen here, where the nominal thickness of 4 nm was modeled as 3.2 nm.

Figure 2b shows the relationship between zero-bias resistance and NiO thickness (solid blue line). We used the extracted fitting parameters from simulation to examine the effects of varying the thickness of NiO from 3 to 6 nm. The results show that contrary

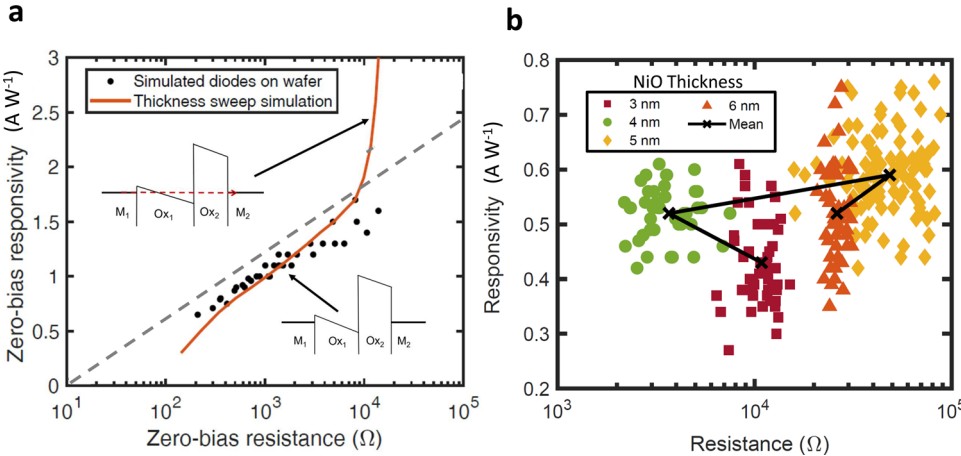

**Fig. 1 Breaking the responsivity/resistance trade-off. a** Responsivity/resistance trade-off at 0 V bias for simulated MI²M diodes with materials set $M_1$/$Ox_1$/$Ox_2$/$M_2$, where $M_1$ and $M_2$ are the respective top and bottom metal electrodes of the diodes, and $Ox_1$ and $Ox_2$ are the two oxides of the diode. These simulated diodes were based on a wafer fabricated and measured by Herner et al., where the spread of diodes across the wafer was marked by the gray dashed line[6]. The solid line depicts the results of varying the thickness of only $Ox_1$. The lower band diagrams illustrates no quasi-bound states at a thickness ratio of 1:1 of $Ox_1$ to $Ox_2$ and the upper one shows a quasi-bound state, marked by a dashed arrow, at a ratio of 1.7:1. **b** Measured responsivity vs. resistance thickness observed for Ni/NiO/Al₂O₃/Cr/Au diodes of different NiO thicknesses. Four batches of lumped-element MI²M diodes were fabricated with varying NiO thickness (3–6 nm in steps of 1 nm), while maintaining the nominal thickness of Al₂O₃ at 1.3 nm.

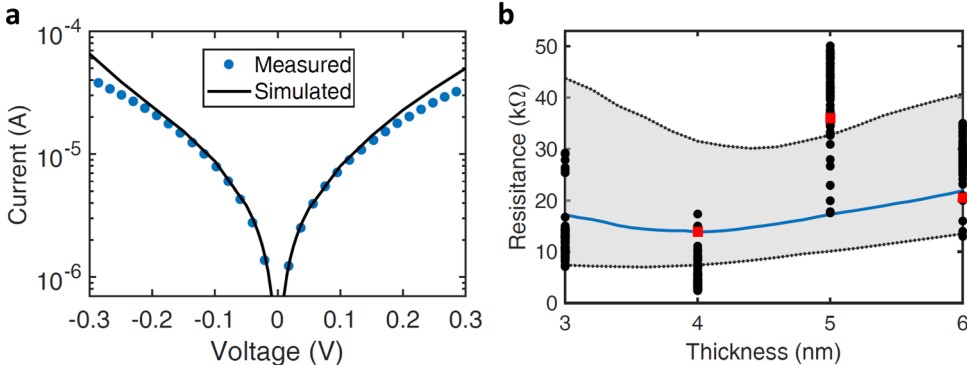

**Fig. 2 Ni/NiO/Al₂O₃/Cr/Au diode simulation analysis. a** Measured (blue filled circles) and simulated (solid black line) $I(V)$ characteristics for 4 nm NiO diode. **b** Simulated resistance vs. NiO thickness trend (solid blue line) with measured diodes at four different thicknesses (solid black circles). The gray shaded area represents simulated thickness variation of ±2 Å in Al₂O₃ thickness. The red squares represent the three diodes that were measured optically under 10.6 μm laser illumination.

to the expected increase in resistance with thickness, a drop in resistance in observed from 3 to 4 nm. Figure 2b also includes measured diodes from Fig. 1b. The gray region shows the simulated ±20% thickness variation of Al₂O₃ across the wafer. That is a thickness variation of ±2 Å for an Al₂O₃ total thickness of 8–12 Å. Resistance variation is also attributed to differences in junction areas of 10% seen in SEM measurements across wafer pieces. What these simulations were unable to explain through Al₂O₃ thickness and area variations is the spread in resistance data of each of the 4 and 5 nm thicknesses, where the 4 and 5 nm NiO diodes exhibited lower and higher resistance than expected from simulations, respectively. The spread in resistance values is attributed to a combination of thickness sensitive interfacial layer properties[18,19] and thickness-dependent NiO chemical composition (discussed in Supplementary Note 3).

**Rectification enhancement at 10.6 μm.** We performed illuminated measurements at 10.6 μm (28.3 THz), as shown in Fig. 3, to explore the possibility of extending the rectification enhancement effects to high frequency. We used a linearly polarized CO₂ laser at 10.6 μm to measure open-circuit voltage ($V_{oc}$) and short-circuit current ($I_{sc}$) of fabricated rectenna structures, as seen in the optical measurement system shown in Fig. 3a. We measured three Ni/NiO/Al₂O₃/Cr/Au devices (with 4 nm NiO, 5 nm NiO, and 6 nm NiO thicknesses). As seen in Fig. 3b, we obtained a cosine-squared relationship for $V_{oc}$ against polarization angle, confirming the response is due to absorption by the antenna. This high-frequency response confirms that the observed responsivity and resistance relationship is not related to electron mobility or interfacial issues. The measured $V_{oc}$ and $I_{sc}$ values for the Ni/4 nm NiO/1 nm Al₂O₃/Cr/Au diode surpass the values for other two device structures, as summarized in Table 1, as well as every reported value in literature[5,20–24].

We use a clamping circuit model of the rectenna configuration to calculate the illuminated DC resistance ($R_0^{Illum}$), defined as the ratio of $V_{oc}$ and $I_{sc}$. This is correct under the assumption that the $I(V)$ curve is linear around the self-bias voltage. In MI²M diodes and at 10.6 μm, $R_0^{Illum}$ is expected to increase compared to zero-bias DC resistance ($R_0^{DC}$) because the voltage division across insulators in an MI²M structure changes from resistive in DC to capacitive at infrared frequencies[24]. Due to the smaller real part of

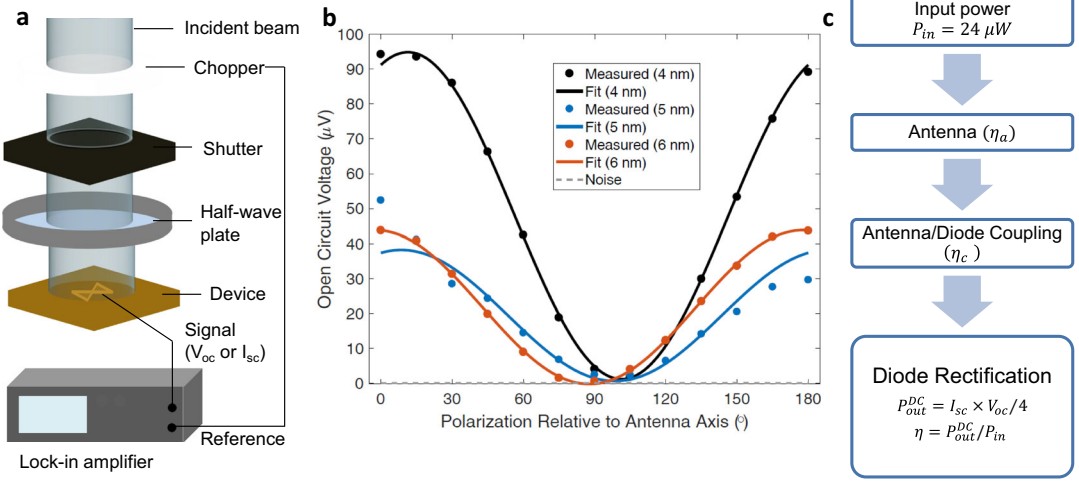

**Fig. 3 Optical measurements and analysis at 28 THz. a** Infrared optical measurement setup with linearly polarized $CO_2$ laser with a maximum beam intensity of 1 W/mm². **b** Open circuit voltage as a function of polarization angle for nominal NiO thicknesses of 4, 5, and 6 nm in Ni/NiO/Al$_2$O$_3$/Cr/Au rectenna structures where 0° and 180° correspond to alignment with the antenna polarization. The three diodes measured here are marked by red squares in Fig. 2b. **c** Waterfall analysis to estimate DC power out. Since the area of the antenna is 24 μm², the input power ($P_{in}$) is calculated to be 24 μW. Figure was created by A. Belkadi, with measured data for **b** provided by A. Weerakkody.

---

| Table 1 Summary of analysis of optical measurements for the Ni/NiO/Al$_2$O$_3$/CrAu rectenna at 10.6 μm wavelength. | | | | | | |
|---|---|---|---|---|---|---|
| $t_{NiO}$ (nm) | $V_{oc}$ ($\mu$V) | $I_{sc}$ (nA) | $R_0^{DC}$ (k$\Omega$) | $R_0^{DC-Illum}$ (k$\Omega$) | $\eta_c$ (%) | $\eta$ $(10^{-9})$ |
| ine 4 | 91.7 | 14.3 | 13.9 | 5.2 | 5.1 | 17 |
| 5 | 41 | 0.6 | 35.8 | 66.9 | 0.4 | 0.26 |
| 6 | 43.8 | 2.1 | 21.8 | 19.6 | 1.4 | 1.1 |

The table includes NiO thickness ($t_{NiO}$), measured open-circuit voltage ($V_{oc}$) and short-circuit current ($I_{sc}$), differential resistance from the DC$I(V)$ ($R_0^{DC}$) and high-frequency differential resistance ($R_0^{DC-Illum}$) and calculated coupling efficiency ($\eta_c$) and power conversion efficiency $\eta$ for three diodes (see Supplementary Note 4).

the complex dielectric constant of Al$_2$O$_3$ compared to NiO, more voltage is applied across Al$_2$O$_3$ resulting in less band-bending in the low barrier material (NiO). With less bending in NiO, electrons tunnel through a thicker triangular region, which results in higher resistance such that $R_0^{Illum} > R_0^{DC}$, as discussed in detail in Belkadi et al.[24] In the structures reported in this work, we were able to achieve a reduction in $R_0^{Illum}$ compared to $R_0^{DC}$, as presented in Table 1. The improvement in $R_0^{Illum}$ from DC (63% drop) is extended to a higher coupling efficiency ($\eta_c$) and a higher overall efficiency ($\eta$) for the 4 nm structure. A less dramatic improvement $R_0^{Illum}$ from DC (10% drop) is observed in the 6 nm structure. Because the optically measured 5 nm structure does not fall on the same trend line as the 4 and 6 nm structures (Fig. 2b), and due to thickness-dependent effects such as interfacial layers and NiO chemical compositions, we do not observe the same effects in the 5 nm structure. This is supported by the measured increase in $R_0^{Illum}$ compared to $R_0^{DC}$ of the 5 nm structure, and the low coupling efficiency and total conversion efficiency.

## Discussion

When the diode is unbiased in DC, the band structure reaches thermodynamic equilibrium when the Fermi levels of its two metal electrodes are aligned. The tunneling probability of electrons, which depends on the band-bending of the insulators, determines show much current is generated from the structure. Band-bending of

insulators depends on barrier heights at metal/oxide interfaces, conduction band offset at the interface of Ox$_1$ and Ox$_2$, and oxide thicknesses. Resonant tunneling occurs when electrons tunnel through discrete quasi-bound states in a triangular quantum well formed between two oxides, as seen in Fig. 4a. Electrons with energies matched to the energy levels of quasi-bound states in the quantum well (Fig. 4b), can reach the opposite side of the structure with less reflections, thus producing a higher current compared to structures that are not in resonance. In MI$^2$M diodes, thickness ratio variations allow for the modification of $I(V)$ characteristics by altering the shape of the triangular barrier, as shown in Fig. 4a–c. Making the oxide with large electron affinity and low barrier height thicker results in more voltage applied across it, more bending, and a deeper and broader well, thus allowing a quasi-bound state to exist closer to self-bias voltage.

As shown in Fig. 4, a resonant well is formed in the NiO adjacent to the Al$_2$O$_3$ and by increasing the thickness of NiO, the Ox$_1$, the band bending becomes sufficient to form a deep and broad quantum well. Increasing the thickness of the higher electron affinity oxide (NiO) increases the depth and width of the well formed between NiO and Al$_2$O$_3$, and thus allows electrons to reach and tunnel through quasi-bound states in the well, as shown in Fig. 4b. When the quasi-bound state is sufficiently far from the Fermi level of M$_1$, the electrons tunnel through a portion of Ox$_1$, and then drift or ballistically traverse at or above the conduction band edge of Ox$_1$, as seen in Fig. 4a, c. In reverse bias, electrons tunnel through Ox$_1$ (Al$_2$O$_3$) only for all three structures. In forward bias, electrons tunnel through Ox$_2$ and a decreasing Ox$_1$ triangular region with the increase in thickness. As thickness increases, responsivity, defined as the asymmetry from forward and reverse biasing the structures, increases as well. From simulations in Fig. 1a, we were able to obtain the band diagram properties necessary to observe resonant tunneling behavior: the low barrier height dielectric (with large electron affinity, Ox$_1$) should be thicker than the high barrier dielectric (with small electron affinity, Ox$_2$), fostering a well in between. Varying the thickness of only Ox$_1$ allows the dimensions of the quantum well to change, which enable us to engineer the band line-up to achieve resonant tunneling closer to the self-bias voltage. Following these design rules, we were able to observe resonant

**Fig. 4 Energy-band diagrams of an MI²M structure with varying Ox₁:Ox₂ ratios. a** 1:1, **b** 2:1, and **c** 3:1. The dashed line represents electron tunneling from the Fermi level of $M_1$ to $M_2$. In **a** and **c**, Fowler–Nordheim tunneling occurs through $Ox_1$ and direct tunneling through $Ox_2$. In **b**, electrons tunnel though the quasi-bound state present in the triangular quantum well, which enhances tunneling probability, increases current and responsivity and decreases resistance.

tunneling effects experimentally with Ni/NiO/Al₂O₃/Cr/Au diodes (Fig. 1b). This unprecedented experimental relationship provides a clear demonstration of resonant tunneling effects.

We support this conclusion by carrying out a simulation analysis of fabricated diodes of different NiO thicknesses (Fig. 2) as well as high-frequency optical measurements (Fig. 3). For structures with resonant tunneling effects, a higher total conversion efficiency is observed. This is because in AC, we expect the Fermi level ($E_F$) of Ni to approach the quasi-bound state more closely than in DC[24], thus improving the tunneling probability drastically, as the tunneling current depends exponentially on the energy difference between the energy state and the $E_F$. The improvement in tunneling probability due to resonant tunneling effects is reflected in a reduction in $R_0^{Illum}$ for the 4 nm NiO structure, as seen in Table 1, as opposed to an increase in $R_0^{DC-Illum}$ for the 5 nm NiO structure due to material-driven voltage division. In the 6 nm NiO structure, the drop in $R_0^{DC-Illum}$ is less dramatic (10%) than the 4 nm NiO structure (63%) since its quasi-bound state is further away from the Fermi level of Ni compared to the 4 nm NiO structure (discussed in Supplementary Fig. 2). Tunneling probabilities for the 4, 5 and 6 nm structures and the location of quasi-bound states in the triangular quantum well are discussed in Supplementary Note 1.

In this work, we experimentally demonstrate resonant tunneling effects in Ni/NiO/Al₂O₃/Cr/Au MI²M diodes, where a reduction in resistance is observed with an increase in responsivity and nonlinearity. The change in NiO thickness in MI²M diodes allows for the modification of the $I(V)$ characteristics by altering the depth and width of the quantum well formed between the two oxides so that the metal Fermi level is aligned with quasi-bound states in the well. Additionally, the well depth can be increased through biasing the diode at higher voltages or through changes in oxide voltage division due to capacitive voltage division at high frequency. We find that the reason resonant tunneling has been hard to demonstrate in the class of MI²M diodes is the difficulty of building low-barrier diodes with wells that are sufficiently deep and wide at low operating voltages to accommodate bound states, and fabrication limitations such as sub-nm control of thicknesses. We believe this is why this behavior of decreasing resistance with increasing responsivity has not been observed in any of our other MI²M diode material combinations such as Co₃O₄/TiO₂, NiO/Nb₂O₅, or NiO/TiO₂. Deep wells are necessary to reach quasi-bound states at low voltages necessary for energy harvesting applications. These NiO/Al₂O₃ MI²M diode rectennas demonstrate record-setting improved total conversion efficiency, beating the previous state-of-the-art by a factor of a 100. These results open a path towards efficient MIM-based optical rectennas for waste heat harvesting and thermoradiative systems[25–28].

## Methods

**Device fabrication**. A shadow mask process, which allows for a single self-aligned mask layer, is used to fabricate MIM devices with small feature sizes on the order of 100 nm. We start with a silicon wafer with a 300 nm layer od thermally grown SiO₂[23]. We spin polymethyl methacrylate (PMMA) in a 4% anisole solution onto the wafer to a thickness of 260 nm and coat the surface with 60 nm of evaporated germanium. We pattern the surface with an ASML 5500 248 nm DUV stepper. We etch the pattern into the germanium with a CF₄ etch and remove the underlayer of PMMA with a O₂ plasma clean. The O₂ plasma is run at a relatively high pressure (≈700 mT) to ensure the PMMA removal undercuts the Ge by at least 0.5 μm. The metals and dielectrics were deposited on a substrate by thermal evaporation and sputtering, respectively. Metal 1, the first metal layer in the MIM stack, is a 35 nm layer of Ni evaporated at an 43° from the right, and Metal 2 is 2.6 nm of Cr layer followed by 46 nm of Au evaporated at normal incidence. All the metal evaporations were done at a rate of 0.2 nm/s and the chamber base pressure was $4 \times 10^{-6}$ Torr. The oxide stack consists of NiO as the dielectric adjacent to Ni and Al₂O₃ as the second oxide. NiO was deposited by DC reactive sputtering at a power of 60 W with 30 SCCM of oxygen and 20 SCCM of argon. We deposited 1 nm of Al₂O₃ by RF sputtering using an Al₂O₃ target at 75 W with 50 SCCM of argon. Both oxide depositions were done at $2.5 \times 10^{-3}$ Torr. Three 4 in. wafers were used for this study. Each wafer was divided into four quadrants, with each quadrant representing a different NiO thickness. The order in which NiO thicknesses were deposited on wafer quadrants was varied to eliminate memory effects. For example, the first wafer quadrants had NiO deposited in the following order: 3, 4, 5, and 6 nm while the second wafer NiO deposition sequence was: 6, 5, 4, and 3 nm.

**Physical characterization**. For characterization, three samples were prepared with thick Al₂O₃ (20 nm), thick NiO (30 nm) on thick Al₂O₃ and thin NiO (2 nm) on thick Al₂O₃. Wafers with thick dielectrics were considered as the bulk samples whereas the wafer with thin NiO was considered as the interfacial sample. X-ray photoelectron spectroscopy (XPS) was done on these samples to measure the chemical composition and valence band offset (VBO) at the interface of NiO and Al₂O₃. Samples were measured at a 90° take-off-angle yielding a penetration depth of <10 nm. The scanning area was 500 μm in diameter and measurements were performed with a Kratos Axis HSi with a monochromatic Al kα x-ray source. Charge neutralization of the sample surface was achieved by a low-energy electron flood gun. We used a pass energy of 160 eV to ascertain survey spectra and a pass energy of 40 eV was used to perform high-resolution core level spectra. These samples were also used to obtain optical properties and bandgap by UV/visible/near-IR variable angle spectroscopic ellipsometry (VASE). We combined XPS and VASE results to generate the band line-up.

**Electrical and optical measurements**. Once fabrication was complete, DC measurements were performed to obtain the diode's $I(V)$ characteristics in DC. We used a four-point probe configuration to perform $I(V)$ measurements so that the active diode junction could be isolated from the parasitic resistances such as lead resistance. A Keithley 2612 source meter was used to source a voltage across two pads and a HP 3478A digital multimeter was used to measure the voltage drop across the junction. We used a mercury switch to short out all the four contact pads during probe manipulation to prevent static discharge from damaging the MIM junction. The rectenna was illuminated with 10.6 μm linearly polarized radiation from a pulsed Synrad 48-1SWJ CO₂ laser. The laser source was pulse width modulated by Agilent 3220A function generator at 20 kHz. The noise level under dark conditions was determined by having the laser beam pass through a ThorLabs SH05 shutter. A half-wave plate (ThorLabs PRM1Z8) was used in the optical path to rotate the laser polarization with respect to the antenna axis. Rectified voltage and/or current responses were measured by a lock-in amplifier (SRS830) and the reference signal for the lock-in amplifier was generated by a mechanical chopper at 1.8 kHz.

**Calculation of tunneling current**. We used a quantum mechanical simulator based on Simmons' generalized formula[29] to calculate tunneling currents and model the experimental data in this work. Tunneling current density $J(V)$, under the assumption that effective masses in each metal region are equal to the electron rest mass ($m_L = m_R = m_0$), can be written as

$$J(V) = \frac{4\pi m_0 q}{h^3} \int_0^\infty T(E_x) dE_x \int_{E_x}^\infty \{f_L(E) - f_R(E + qV)\} dE \quad (1)$$

where $V$ is the voltage applied across the diode, $q$ is electron charge, $h$ is Planck constant, $E$ is the tunneling electron energy and $f_L$ and $f_R$ are the Fermi-Dirac

distribution function in the left and right metal electrodes, respectively. The tunneling probability $T(E_x)$ is calculated using a transfer-matrix method to find the plane-wave solution for the Schrödinger equation, such that

$$T(E_x) = \frac{k_{N+1}}{k_0} \frac{|A_{N+1}^+|^2}{|A_0^+|^2} \tag{2}$$

where $k_0 = \sqrt{2m_e m_L q E_x}/\hbar$, $k_{N+1} = \sqrt{2m_e m_R q E_x}/\hbar$, $m_L$ and $m_R$ are mass of electron in left metal (cathode) and right metal (anode), respectively, $|A_0^+|^2$ is amplitude of incoming wave and $|A_{N+1}^+|^2$ is amplitude of transmitted wave[8].

## Data availability
Data are available from the corresponding author upon reasonable request.

## Code availability
Codes for the quantum tunneling simulator are available from the corresponding author upon request.

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

## Acknowledgements
The authors would like to thank David Doroski for assistance with fabrication of the structures and John Stearns for valuable discussions and helpful comments. This work was carried out in part under contract from RedWave Energy Inc. and funded in part by the Advanced Research Projects Agency—Energy (ARPA-E), U.S. Department of Energy, under Award Number DE-AR0000676. A portion of the fabrication was carried out at UCSB Nanofabrication Facility. G. Moddel holds stock in RedWave Energy, Inc.

## Author contributions
A.B. and A.W. contributed equally to this work and conducted the resonant tunneling diode study under the supervision of G.M. A.W. fabricated the devices, characterized materials and measured the rectenna response of all devices. A.B. performed device modeling and analyzed the experimental data. All authors participated in the discussion and data interpretation. A.B. drafted the manuscript. All authors contributed to the final version of the manuscript.

## Competing interests
G.M. holds stock in RedWave Energy, Inc.
