## [Peer Review File · Nature Communications]

Reviewers' Comments:

Reviewer #1:

Remarks to the Author:

Introduction:

The presented paper provides an insight into the operation of double-insulator tunnelling diodes (MIIM) from a more industrial point of view. The novel contribution of the paper to the field is based on:

- The work is aimed at better understanding of the adopted theory and the artifacts that accompany the tunnelling diode industrial process.
- Discovery, for a certain insulator thickness range, of an inverse relationship between the insulator thickness and the zero-bias resistance R_0^{DC}
- A parametric sweep research is conducted to analyse the impact of the design parameters on the overall performance in a yield-analysis-like study.

According to the journal regulations, the article features the same structure as other published materials, so a brief introduction is presented first followed by a demonstration of the obtained results for different scenarios. In the end, a discussion is demonstrated and the utilized methods in the study are showcased.

Although the contribution of the paper is worth publishing, there are, however, some conceptual issues that need to be addressed by the authors to consummate the work in the final version.

The formatting and language is okay with just minor misprints which are mentioned in this letter.

Conceptual Issues:

Abstract:

You mentioned that you “achieve the usually mutually exclusive desired characteristics of low resistance ($R_{0DC} \sim 13 \text{ k}\Omega$ for $0.035 \text{ }\mu\text{m}^2$) and high responsivity ($\beta_0 = 0.5 \text{ A/W}$)” with conversion efficiency more than 100 times more than the state-of-the-art energy harvesting rectennas. In [1], for example, the writers set the figures of merit that can be considered for an efficient rectenna -or rectifier-. That is, asymmetry, nonlinearity, responsivity, and dynamic resistance. Also, the results mentioned in Hemour et al [2], correspond to the set values. Comparing these values with what you have obtained, which I agree is outstanding, gives an indication that there is still way to improve. For the reader I think it is good to mention some comparison with more efficient commercial devices, even in the lower GHz range just to give a good idea where the tunnelling devices are placed.

Simulation-based resonant tunneling analysis:

In p. 2, Col. 2, L. 16, you mention “These measured values were used as a starting point for the simulations, and were varied...”referring to the diode cross section area and the nominal oxide layer thickness where these values have been measured using (SEM), and (VASE) methods respectively. My questions here are:

1. For this area, is there a variation among the fabricated devices? How much is that?
2. For the thickness, a similar statistical analysis of the variation is quite beneficial (just for the measured and not for the simulated which you have already done).
3. For the interface between the oxides, the surface metal roughness is considered very important as it could give rise to other forms of tunnelling (i.e. a smoother surface gives closer results to theoretical study output). Also here, an indication of the surface roughness is useful.
4. The variation of effective mass, electron affinity, and work function values (even for the same oxides and metals but for different junctions and crystallization forms, plays a vital role in defining the current values) if you could provide a table to compare your values with other results from the literature?

Rectification enhancement at 10.6 μm :

In p.4, Col.2, L.22 regarding the discussion of the voltage division difference between AC and DC: This point is unclear and needs more explanation.

For the use of the dielectric constant, I think they have been used just for the calculation of the image forces lowering. If you are talking about typical voltage division techniques calculation for MIIM diodes as the one mentioned in [3], then I think the electric field density D continuity at the boundary condition is true down to the DC!

On the other hand, if you are talking about this point based on the "bypassing" of the tunnelling structure that happens at higher frequencies due to the reduced capacitor reactance, which results in poor rectification for the whole system -simply because much more current passes in the linear capacitive part than the nonlinear part-, then it needs to be explained in a different way to not get it wrong.

A mathematical model how you find RDC is highly beneficial to support what you say both in your paper [22] and in this paper.

Format and Language:

- In p.5: Figure 3: Seems to not be cross referenced in the text.
- Consistency of figures cross referencing: Sometimes you use Fig. (e.g. p.5, Col.1, L.10, Fig. 2b), and sometimes you use Figure. (e.g. p.5, Col.2, L.3, Figure 4a.). Is it different where you mention the cross referencing? (i.e. between the results and the discussion sections?)
- Some mistyping as in the table below:

Position	Error	Correction
P.3, Col.2, L.9	tohlabsorption	Probably: to the absorption
P.6, Figure. 4 caption	(c) 1:1 (d) 2:1 and (c) 3:1	Probably: (a) 1:1 (b) 2:1 and (c) 3:1?
P.2, Column.1	A conditioning circuit ... need to be used	needs to be used
P.2, Column.1	ADS LineCal operator is ... and optimized	and optimize
P.2, Column.2	were transfer into ADS	were transferred into ADS

References:

- [1] E. Donchev, J. S. Pang, P. M. Gammon, A. Centeno, F. Xie, P. K. Petrov, J. D. Breeze, M. P. Ryan, D. J. Riley, and N. M. Alford, "The rectenna device: From theory to practice (a review)", *MRS Energy & Sustainability-A Review Journal*, vol. 1, 2014.
- [2] S. Hemour and K. Wu, "Radio-frequency rectifier for electromagnetic energy harvesting: Development path and future outlook", *Proceedings of the IEEE*, vol. 102, no. 11, pp. 1667–1691, Nov. 2014, ISSN: 1558- 2256.
- [3] S. Grover and G. Moddel, "Engineering the current–voltage characteristics of metal–insulator–metal diodes using double-insulator tunnel barriers", *Solid-State Electronics*, vol. 67, no. 1, pp. 94–99, 2012.

Reviewer #2:

Remarks to the Author:

-What are the major claims of the paper?

An improvement in energy conversion efficiency by more than a factor of 100 over the current state-of-the-art using oxide thicknesses to sub-nanometer accuracy.

Are they novel and will they be of interest to others in the community and the wider field?

Yes, but metal-insulator interfaces should have been discussed in depth since the thinner the oxide layers the more electrons would be trapped at these interfaces due to inter-atomic diffusion and electron drift and grain boundary and oxygen vacancy effects that are resulting in less mobility and lower responsivity (i.e., high resistivity due to that the area is in nanometer scale) and non-linearity (anisotropy)!? Moreover, it is recommended to try different metal/oxide/oxide/metal combinations in addition to Ni/NiO/Al₂O₃/Cr/Au to show if this observation is independent of metal and oxides type and amount of layers.

Reviewer #3:

Remarks to the Author:

The authors report improvement of responsivity and conversion efficiency for electromagnetic wave at 30THz using metal-insulator-metal (MI2M) rectifier with antenna structure, which can be reasonably explained by current-voltage characteristics derived from resonant tunneling transport model. The results are important and suggestive. For further understandings, the manuscript should be revised accordance with the following questions and comments.

1)

The basic principle of rectification is the nonlinearity or asymmetry around $V=0$ of the current-voltage characteristics.

Among the possible conduction mechanisms of MI2M, it should be described in the text why resonant tunneling is more advantageous for developing asymmetry in the current-voltage characteristic than other conduction mechanisms. In particular, discuss the advantages comparing to for example Schottky junctions or single barrier tunneling diodes.

2)

Frequency of 30THz used in this study is much higher than cut off frequency ($1/2 \cdot \pi \cdot CR$) of the MI2M diodes. Is it consistent to rectification output of the diode? Equivalent circuit may be helpful to discuss the detail.

3)

Relation between Fig.1(a) and (b) is difficult to understand.

Red solid line in Fig.1(a) exhibits steep increase of responsivity with keeping resistance almost constant under resonant tunneling region. However, in the Fig.1(b), experimental plots distribute in wide range of resistance (10^3 - 10^5 ohm) and significant increase of responsivity can not be seen and the responsivity values seem to be smaller than the value shown in Fig.1(a). Therefore it is difficult to determine the validity of the description that the results plotted in Fig.1(b) is based

on resonant tunneling.

4)

There are some miss-type errors in figure caption of Fig.4.

5)

In Fig.4, mechanism of resonant tunneling is comprehensive and the reason of resistance reduction at resonant tunneling shown in (b) is understandable. However, it is difficult to understand increase of responsivity because origin of responsivity is asymmetry of I-V curve instead of resistance itself. Explanation of the origin of responsivity may be helpful in the text.

REVIEWER COMMENTS

Reviewer #1 (Remarks to the Author):

Introduction

The presented paper provides an insight into the operation of double-insulator tunnelling diodes (MIIM) from a more industrial point of view. The novel contribution of the paper to the field is based on:

- The work is aimed at better understanding of the adopted theory and the artifacts that accompany the tunnelling diode industrial process.
- Discovery, for a certain insulator thickness range, of an inverse relationship between the insulator thickness and the zero-bias resistance R_0^{DC}
- A parametric sweep research is conducted to analyse the impact of the design parameters on the overall performance in a yield-analysis-like study.

According to the journal regulations, the article features the same structure as other published materials, so a brief introduction is presented first followed by a demonstration of the obtained results for different scenarios. In the end, a discussion is demonstrated and the utilized methods in the study are showcased.

Although the contribution of the paper is worth publishing, there are, however, some conceptual issues that need to be addressed by the authors to consummate the work in the final version.

The formatting and language is okay with just minor misprints which are mentioned in this letter.

Response:

We thank the reviewer for their thorough and insightful review of our paper. We have addressed Each of the reviewer's points below and have revised the manuscript to clarify all the points and add the details needed to support them.

Conceptual Issues:

Abstract:

You mentioned that you “achieve the usually mutually exclusive desired characteristics of low resistance ($R_0^{DC} \sim 13 \text{ k}\Omega$ for $0.035 \mu\text{m}^2$) and high responsivity ($\beta_0 = 0.5 \text{ A/W}$)” with conversion efficiency more than 100 times more than the state-of -the-art energy harvesting rectennas. In [1], for example, the writers set the figures of merit that can be considered for an efficient rectenna -or rectifier-. That is, asymmetry, nonlinearity, responsivity, and dynamic resistance. Also, the results mentioned in Hemour et al [2], correspond to the set values. Comparing these values with what you have obtained, which I agree is outstanding, gives an indication that there is still way to improve. For the reader I think it is good to mention some comparison with more efficient commercial devices, even in the lower GHz range just to give a good idea where the tunnelling devices are placed.

Response:

We thank the reviewer for this suggestion and very good references. The mentioned four figures of merit are the ones we used to evaluate our diodes. We focus on responsivity since it is related to asymmetry and nonlinearity. The challenge lies in the dance between responsivity and resistance to achieve the needed cutoff frequency for the desired application. Schottky diode have high responsivities and high resistances that cannot be used for energy harvesting applications with terahertz cutoff frequencies. We agree such a comparison would be good for the audience.

Due to wording limitations in the abstract, we have added the following line to our manuscript in the introduction (p. 1, Col.1) “Schottky diodes have good rectification characteristics with responsivity values up to 5 A/W, but their high resistance results in a cut-off frequency of around 40 GHz.” and cited Donchev wet al.'s and Hemour et al.'s work.

Simulation-based resonant tunneling analysis:

In p. 2, Col. 2, L. 16, you mention “These measured values were used as a starting point for the simulations, and were varied...”referring to the diode cross section area and the nominal oxide layer thickness where these values have been measured using (SEM), and (VASE) methods respectively. My questions here are:

1. For this area, is there a variation among the fabricated devices? How much is that?

Response:

Yes. Devices varied in area between 0.031 μm and 0.037 μm (~11% max variation). We have added that to the text in p.2, Col. 2.

2. For the thickness, a similar statistical analysis of the variation is quite beneficial (just for the measured and not for the simulated which you have already done).

Response:

We agree. With each deposition step for a wafer fabricated, a blank control wafer is included. Oxide thickness in at least 5 positions around the control wafer is measured. On average, the variation is about 2-5 Angstroms for a 4-inch wafer, which is our reference for the simulation variation. We have added that to the text in p.2, Col. 2.

It is very difficult though to measure every spot on the control wafer and to relate that to single fabricated and measured devices. That measured thickness might still not be exact since it could change when depositing more oxides or metals on top.

3. For the interface between the oxides, the surface metal roughness is considered very important as it could give rise to other forms of tunnelling (i.e. a smoother surface gives closer results to theoretical study output). Also here, an indication of the surface roughness is useful.

Response:

Surface roughness is considered here to be very smooth. We have analyzed our results in collaboration with Prof. Conley’s group and determined that our oxides that are grown or deposited using sputtering are smooth. TEM in Fig. 4 in Pelz et al. [4] illustrates this.

4. The variation of effective mass, electron affinity, and work function values (even for the same oxides and metals but for different junctions and crystallization forms, plays a vital role in defining the current values) if you could provide a table to compare your values with other results from the literature?

Response:

A table has been added to supplementary materials to compare our values to literature values.

Table 1: Work function (Φ) values used in simulations versus those reported in literature.

Material	Fitted values	Literature values	References
Ni	$\Phi = 5$ eV	$\Phi_{Ni} = 5.01$ eV	[3]
Cr/Au	$\Phi = 4.47$ eV	$\Phi_{Cr} = 4.5$ eV $\Phi_{Au} = 5.1$ eV	[4] [3]

For the electron affinity of the oxides, NiO has literature values of 1.4-1.5 eV [5], which is different from our 4.7 eV used in the simulation. We are working on better understanding and characterizing that oxide. Our Al_2O_3 on the other hand has been extensively studied in our most recent paper [6]. Our value of 3.45 eV is due to the nonstoichiometric Al_2O_3 [6] compared to a value of 1.4-1.6 eV for thin stoichiometric Al_2O_3 [7,8,9].

Rectification enhancement at 10.6 μm :

In p.4, Col.2, L.22 regarding the discussion of the voltage division difference between AC and DC: This point is unclear and needs more explanation.

Response:

To add more clarity, we have replaced the term $R_0^{\{DC-Illum\}}$ with $R_0^{\{Illum\}}$ for simplicity. We have reworded that section to be:

We use a clamping circuit model of the rectenna configuration to calculate the illuminated DC resistance R_0^{Illum} , defined as the ratio of V_{oc} and I_{sc} . This is correct under the assumption that the $I(V)$ curve is linear around the self-bias voltage. In M^2M diodes and at 10.6 μm , R_0^{Illum} is expected to increase compared to zero-bias DC resistance

(R_0^{DC}) because the voltage division across insulators in an MI²M structure changes from resistive in DC to capacitive at infrared frequencies [24]. Due to the smaller real part of the complex dielectric constant of Al₂O₃ compared to NiO, more voltage is applied across Al₂O₃ resulting in less band-bending in the low barrier material (NiO). With less bending in NiO, electrons tunnel through a thicker triangular region, which results in higher resistance such that $R_0^{illum} > R_0^{DC}$ as discussed in detail in Belkadi et al. [24].

For the use of the dielectric constant, I think they have been used just for the calculation of the image forces lowering. If you are talking about typical voltage division techniques calculation for MIIM diodes as the one mentioned in [3], then I think the electric field density \mathbf{D} continuity at the boundary condition is true down to the DC!

Response:

In [3] referenced here, the dielectric constant is not just used to calculate the image forces lowering but is in fact used to determine the voltage division of the structure using a capacitive method as would be done at high frequency. The simulator works by determining an energy-band profile at a certain bias using the condition for continuity of the electric displacement vector at each insulator interface, which depends on dielectric constants. This is an erroneous method since internal oxide resistances dominate at DC and not capacitive voltage division [6].

On the other hand, if you are talking about this point based on the "bypassing" of the tunnelling structure that happens at higher frequencies due to the reduced capacitor reactance, which results in poor rectification for the whole system -simply because much more current passes in the linear capacitive part than the nonlinear part-, then it needs to be explained in a different way to not get it wrong.

Response:

Though correct, that is not the point we are trying to make here. We simply want to point out one thing: that under the absence of resonant tunneling, resistance typically increases in MI²M structures from DC to under illumination. In this structure, the expected increase is due to the dielectric constants of Al₂O₃ and NiO. We hope rewording that section has helped in making and clarifying our point.

A mathematical model how you find RDC is highly beneficial to support what you say both in your paper [22] and in this paper.

Response:

We thank you for the suggestions. We have added this to our supporting material.

Format and Language:

- In p.5: Figure 3: Seems to not be cross referenced in the text.

Response:

Figure 3 is cross referenced in the section titled "Rectification enhancement at 10.6 μm". Figure 3 includes three subfigures: a, b and c. Figure 3a is referenced on page 4, column 2, line 4. Figure 3b is referenced on page 4, column 2, line 7. For added clarification, we've cross referenced the figure in the first sentence of this section (highlighted on page 4).

- Consistency of figures cross referencing: Sometimes you use Fig. (e.g. p.5, Col.1, L.10, Fig. 2b), and sometimes you use Figure. (e.g. p.5, Col.2, L.3, Figure 4a.). Is it different where you mention the cross referencing? (i.e. between the results and the discussion sections?)

Response:

We thank the reviewer for pointing this out. We have changed everything to Fig. in the manuscript (highlighted).

- Some mistyping as in the table below:

Position	Error	Correction
P.3, Col.2, L.9	tohlabsorption	Probably: to the absorption
P.6, Figure. 4 caption	(c) 1:1 (d) 2:1 and (c) 3:1	Probably: (a) 1:1 (b) 2:1 and (c) 3:1?
P.2, Column.1	A conditioning circuit ... need to be used	needs to be used
P.2, Column.1	ADS LineCal operator is ... and optimized	and optimize
P.2, Column.2	were transfer into ADS	were transferred into ADS

Response:

We thank the reviewer for pointing these errors out. We have made the suggested corrections for points 1 and 2 in the manuscript (highlighted). Points 3, 4 and 5 cannot be found in our manuscript.

References:

- [1] E. Donchev, J. S. Pang, P. M. Gammon, A. Centeno, F. Xie, P. K. Petrov, J. D. Breeze, M. P. Ryan, D. J. Riley, and N. M. Alford, "The rectenna device: From theory to practice (a review)", *MRS Energy & Sustainability-A Review Journal*, vol. 1, 2014.
- [2] S. Hemour and K. Wu, "Radio-frequency rectifier for electromagnetic energy harvesting: Development path and future outlook", *Proceedings of the IEEE*, vol. 102, no. 11, pp. 1667–1691, Nov. 2014, ISSN:1558- 2256.
- [3] S. Grover and G. Moddel, "Engineering the current–voltage characteristics of metal–insulator–metal diodes using double-insulator tunnel barriers", *Solid-State Electronics*, vol. 67, no. 1, pp. 94–99, 2012.
- [4] Pelz, B., and G. Moddel, "Demonstration of distributed capacitance compensation in a metal-insulator-metal infrared rectenna incorporating a traveling-wave diode", *Journal of Applied Physics* 125.23 (2019): 234502.
- [5] Mitrovic, I., Weerakkody, A., Sedghi, N., Ralph, J., Hall, S., Dhanak, V., Luo, Z., and Beeby, S., "Controlled modification of resonant tunneling in metal-insulator-insulator-metal structures", *Appl. Phys. Lett.* 2018, 112, No. 012902.
- [6] Weerakkody, Ayendra, Belkadi, Amina and Moddel, Garret, "Nonstoichiometric Nanolayered Ni/NiO/Al₂O₃/CrAu Metal–Insulator–Metal Infrared Rectenna", *ACS Applied Nano Materials*, 2021.
- [7] Alimardani, N.; King, S. W.; French, B. L.; Tan, C.; Lampert, B. P.; Conley, J. oF., Jr., "Investigation of the impact of insulator material on the performance of dissimilar electrode metal-insulator-metal diodes", *J. Appl. Phys.* 2014, 116, No. 024508.
- [8] Taylor, T. R.; Hansen, P. J.; Pervez, N.; Acikel, B.; York, R. A.; Speck, J. S., "Influence of stoichiometry on the dielectric properties of sputtered strontium titanate thin films", *J. Appl. Phys.* 2003, 94, 3390– 3396.
- [9] Koffyberg, F. P., and F. A. Benko, "p-Type NiO as a Photoelectrolysis Cathod", *Journal of the electrochemical society* 128.11 (1981): 2476.

Reviewer #2 (Remarks to the Author):

-What are the major claims of the paper?

An improvement in energy conversion efficiency by more than a factor of 100 over the current state-of-the-art using oxide thicknesses to sub-nanometer accuracy.

Are they novel and will they be of interest to others in the community and the wider field?

[Comment] Yes, but metal-insulator interfaces should have been discussed in depth since the thinner the oxide layers the more electrons would be trapped at these interfaces due to inter-atomic diffusion and electron drift and grain boundary and oxygen vacancy effects that are resulting in less mobility and lower responsivity (i.e., high resistivity due to that the area is in nanometer scale) and non-linearity (anisotropy)!?

Response:

We thank the reviewer for the comments. We agree with the reviewer that the thin nature of our metals and insulators creates non-ideal interfacial layers. If these anomalies were present in our structures, electron mobility of the electrons that traverse through NiO/Al₂O₃ layers would be reduced because the carrier transport would be dominated by bulk conduction (Poole-Frenkel). In other words, carrier transit time would not be in the order of femtoseconds as we would expect for quantum mechanical tunneling. One key requirement for a MIM based rectenna structure that operates at optical frequencies is to have a sufficiently fast (transit times must be in femtoseconds) carrier transport mechanism. In this work, we verified the high frequency operation of Ni/NiO/Al₂O₃/Cr/Au rectenna devices at 28.3 THz. Therefore, we can determine these interfacial and material defects are not present in our structures.

Unfortunately, because our metals are reactive and therefore pose issues when attempting to perform surface-sensitive measurements such as XPS, UPS and IPES which require pure metallic films to acquire clean core level information. Our reactive metals form native oxides that would lead to wrong conclusions about the whole MIM stack in the analysis process. Although we acknowledge that issue, we had to take an experimental approach that relies on predictions by a quantum tunneling simulator. To verify whether these interfacial issues were the reason for the improvement in responsivity and nonlinearity, we extended our analysis to 28.3 THz..

We have added the following sentence in the manuscript on page 4, column 2:

This high frequency response confirms that the observed responsivity and resistance relationship is not related to electron mobility or interfacial issues.

[Comment] Moreover, it is recommended to try different metal/oxide/oxide/metal combinations in addition to Ni/NiO/Al₂O₃/Cr/Au to show if this observation is independent of metal and oxides type and amount of layers.

Response:

Based on our experience, Ni/NiO and Nb/Nb₂O₅ yield small barrier heights. Since our GSM fabrication process requires angled and directional evaporations, Nb was not an option (its melting point is too high for thermal evaporations). Therefore, we chose NiO because of the small barrier (~0.1 eV) it creates with Ni. This is important to achieve a small differential resistance and also to push resonant tunneling towards zero-bias [1-3]. If we chose a different interface, a large bias would need to be applied to get the metal Fermi level aligned with a bound state in the quantum well [1-5]. Many groups have worked on metal-insulator-metal diode technology with various material combinations, aiming at thermal energy harvesting. One common factor among most of MIM diode researchers is using Al₂O₃ as a dielectric in their material stack [Al₂O₃ paper, table 1]. This is solely due to its small

dielectric constant of 0.8 at 28.3 THz, leading to a low capacitance and the possibility of high-frequency operation. Most of the high frequency friendly uncompensated rectennas were fabricated with Al₂O₃ [6, 8-10].

Without going into the details of other materials tested, we have added the following sentence in the manuscript on page 6, column 2:

We believe this is why this behavior of decreasing resistance with increasing responsivity has not been observed in any of our other MI²M diode material combinations such as Co₃O₄/TiO₂, NiO/Nb₂O₅ or NiO/TiO₂.

References

- [1] P. Maraghechi, A. Foroughi-Abari, K. Cadien, and A. Y. Elezzabi. Observation of resonant tunneling phenomenon in metal-insulator-insulator-insulator-metal electron tunnel devices. *Applied Physics Letters*, 100(11):113503, 2012.
- [2] A.D. Weerakkody, N. Sedghi, I.Z. Mitrovic, H. van Zalinge, I. Nembr Nouredine, S. Hall, J.S. Wrench, P.R. Chalker, L.J. Phillips, R. Treharne, and K. Durose. Enhanced low voltage nonlinearity in resonant tunneling metal-insulator-insulator-metal nanostructures. *Microelectronic Engineering*, 147:298 - 301, 2015.
- [3] Mitrovic, I., Weerakkody, A., Sedghi, N., Ralph, J., Hall, S., Dhanak, V., Luo, Z., and Beeby, S. Controlled modification of resonant tunneling in metal-insulator-insulator-metal structures. *Appl. Phys. Lett.* 2018, 112, No. 012902.
- [4] Blake J Eliasson, and Garret Moddel. Metal-oxide electron tunneling device for solar energy conversion, March 18 2003. US Patent 6,534,784.
- [5] Sachit Grover and Garret Moddel. Engineering the current-voltage characteristics of metal-insulator- metal diodes using double-insulator tunnel barriers. *Solid-State Electronics*, 67(1):94-99, 2012.
- [6] Weerakkody, Ayendra, Belkadi, Amina and Moddel, Garret, "Nonstoichiometric Nanolayered Ni/NiO/Al₂O₃/CrAu Metal-Insulator-Metal Infrared Rectenna", *ACS Applied Nano Materials*, 2021.
- [7] Bradley Pelz and Garret Moddel. Demonstration of distributed capacitance compensation in a metal-insulator-metal infrared rectenna incorporating a traveling-wave diode at 10.6 μ m. *Journal of Applied Physics*, 2019.
- [8] Jayaswal, Gaurav, et al. Optical rectification through an Al₂O₃ based MIM passive rectenna at 28.3 THz. *Materials Today Energy* 7 (2018): 1-9.].
- [9] Jeffrey A. Bean, Badri Tiwari, Gary H. Bernstein, P. Fay, and Wolfgang Porod. Thermal infrared detection using dipole antenna-coupled metal-oxide-metal diodes. *Journal of Vacuum Science & Technology B: Microelectronics and Nanometer Structures Processing, Measurement, and Phenomena*, 27(1):11-14, 2009.
- [10] Amina Belkadi, Ayendra Weerakkody, and Garret Moddel. Large errors from assuming equivalent dc and high-frequency electrical characteristics in metal-multiple-insulator-metal diodes. *ACS Photonics*, 5(12):4776-4780, 2018.

Reviewer #3 (Remarks to the Author):

The authors reports improvement of responsivity and conversion efficiency for electromagnetic wave at 30THz using metal-insulator-metal (MI2M) rectifier with antenna structure, which can be reasonably explained by current-voltage characteristics derived from resonant tunneling transport model. The results are important and suggestive. For further understandings, the manuscript should be revised accordance with the following questions and comments.

1) The basic principle of rectification is the nonlinearity or asymmetry around $V=0$ of the current-voltage characteristics.

Among the possible conduction mechanisms of MI^2M , it should be described in the text why resonant tunneling is more advantageous for developing asymmetry in the current-voltage characteristic than other conduction mechanisms. In particular, discuss the advantages comparing to for example Schottky junctions or single barrier tunneling diodes.

Response:

We thank the reviewer for the comments. Reasons why resonant tunneling is advantageous are covered in detail in references 7-10. We have added the sentence "Schottky diodes have good rectification characteristics with responsivity values up to 5 A/W, but their high resistance results in a cut-off frequency of around 40 GHz" to highlight that better diodes exist but are limited to low frequency operations which does not work for our target application of energy harvesting.

2) Frequency of 30THz used in this study is much higher than cut off frequency ($1/2 * \pi * CR$) of the MI^2M diodes. Is it consistent to rectification output of the diode? Equivalent circuit may be helpful to discuss the detail.

Response:

*What matters here is the frequency of the whole system not just the diode on its own – that is the cutoff frequency of the antenna, the diode, and a compensation structure, as seen in the diagram below, and discussed in Jayaswal et al. [Jayaswal, Gaurav, et al. "Optical rectification through an Al_2O_3 based MIM passive rectenna at 28.3 THz." *Materials Today Energy* 7 (2018): 1-9.]. We are unfortunately limited by space in the manuscript and have chosen to focus on diode requirements of the system which can be simplified to low resistance and high responsivity. For our systems, we use a bowtie antenna with a resistance R_A . Our diode's cutoff here is around 1 THz.*

3) Relation between Fig.1(a) and (b) is difficult to understand.

Red solid line in Fig.1(a) exhibits steep increase of responsivity with keeping resistance almost constant under resonant tunneling region. However, in the Fig.1(b), experimental plots distributes in wide range of resistance (10^3 - 10^5 ohm) and significant increase of responsivity can not be seen and the responsivity values seems to be smaller than the value shown in Fig.1(a). Therefore it is difficult to determine the validity of the description that the results plotted in Fig.1(b) is based on resonant tunneling.

Response:

What is presented here in figures 1(a) and 1(b) is simulations vs. fabrication. The solid red line in Fig. 1(a) represents simulation results from fitted parameters. The sweep there is done in steps of 0.1 nm. This served as a prediction that resonant tunneling can be observed in MI^2M structures if the right conditions are met. Fig. 1(b) presents the fabricated NiO/Al_2O_3 results for one material set where the thickness is swept in steps of 1 nm. With

fabricated devices, it is much more difficult to step in 0.1 nm (almost impossible). It is also very difficult to find/deposit material with exact material properties as simulations. What we have discovered from our experimental work with these oxides is that certain interface states are present at specific oxide ratios, such as the 5 nm NiO structures (yellow diamonds). What can be seen in Fig. 1(b) is that resistance decreases with an increase in responsivity from 3 nm to 4 nm, which is in agreement with the trend in Fig. 1(a). We have carried out further simulations and measurements (illuminated measurements under 10.6 μm and VASE and XPS) to verify our hypothesis as can be seen in Fig. 2 and Fig. 3. Fig. 1(b) served as the first observation of a nonlinear resistance/responsivity relationship and of the possibility that something interesting might be happening here, especially that it was repeated three times.

4) There are some miss-type errors in figure caption of Fig.4.

Response:

We thank the reviewer for his comment. We have fixed that typo.

5) In Fig.4, mechanism of resonant tunneling is comprehensive and the reason of resistance reduction at resonant tunneling shown in (b) is understandable. However, it is difficult to understand increase of responsivity because origin of responsivity is asymmetry of I-V curve instead of resistance itself. Explanation of the origin of responsivity may be helpful in the text.

Response:

The responsivity is defined as the asymmetry from forward and reverse biasing the structures. In Fig. 4, this can be seen by forward biasing and reverse biasing each structure and looking at the ratio of the areas underneath the band diagram. Increasing NiO thickness would result in a higher asymmetry because of more bending with bias. An example is included below.

We have added the following sentence in the manuscript on page 6, column 1:

In reverse bias, electrons tunnel through Ox_1 (Al_2O_3) only for all three structures. In forward bias, electrons tunnel through Ox_2 and a decreasing Ox_1 triangular region with the increase in thickness. As thickness increases, responsivity, defined as the asymmetry from forward and reverse biasing the structures, increases as well.

Reviewers' Comments:

Reviewer #1:

None

Reviewer #2:

None

Reviewer #3:

Remarks to the Author:

The authors have responded appropriately to the reviewer's questions and comments and have made revisions to the manuscript.

Especially a comment of comparison between Schottky diodes is comprehensive and helpful for better understandings of originality of this work.

It is hoped that the difference between the experimental and theoretical results will be filled in by future research.

REVIEWER COMMENTS

Reviewer #3 (Remarks to the Author):

The authors have responded appropriately to the reviewer's questions and comments and have made revisions to the manuscript. Especially a comment of comparison between Schottky diodes is comprehensive and helpful for better understandings of originality of this work. It is hoped that the difference between the experimental and theoretical results will be filled in by future research.

Response:

We thank the reviewer for their thorough and insightful review of our paper. We hope to fulfil the gap between experimental and theoretical results with more measurements and simulations.